
**Improvement of RAMS precipitation forecast at the short range through lightning data**
**assimilation**
Stefano Federico[1], Marco Petracca[1], Giulia Panegrossi[1], Stefano Dietrich[1]
[1] *ISAC-CNR, UOS of Rome, via del Fosso del Cavaliere 100, 00133-Rome, Italy*
Phone: +390649934209
Fax: +390645488291
s.federico@isac.cnr.it
marco.petracca@artov.isac.cnr.it
g.panegrossi@isac.cnr.it
s.dietrich@isac.cnr.it
www.isac.cnr.it
**Abstract**
This study shows the application of a total lightning data assimilation technique to the RAMS
(Regional Atmospheric Modeling System) forecast. The method, which can be used at high
horizontal resolution, helps to initiate convection whenever flashes are observed by adding water
vapour to the model grid column. The water vapour is added as a function of the flash rate, local
temperature and graupel mixing ratio.  The methodology is set-up to improve the short-term (3h)
precipitation forecast and can be used in real-time forecasting applications. However, results are
also presented for the daily precipitation for comparison with other studies.
The methodology is applied to twenty cases occurred in fall 2012, that were characterized by
widespread convection and lightning activity. For these cases a detailed dataset of hourly
precipitation containing thousands of raingauges over Italy, which is the target of this study, is
available through the HyMeX (HYdrological cycle in the Mediterranean Experiment) initiative.
This dataset gives the unique opportunity to verify the precipitation forecast at the short range (3h)
and over a wide area (Italy).
Results for the 27 October case study show how the methodology works and its positive impact on
the 3h precipitation forecast. In particular, the model represents better the convection over the sea
using the lightning data assimilation and, when convection is advected over the land, the
precipitation forecast improves over the land. It is also shown that the precise location of the
convection by lightning data assimilation, improves the precipitation forecast at fine scales (meso-
$\beta$).





The application of the methodology to twenty cases gives a statistically robust evaluation of the
impact of the total lightning data assimilation on the model performance. Results show an
improvement of all statistical scores, with the exception of the Bias. The Probability of Detection
(POD) increases by 3-5% for the 3h forecast and by more than 5% for daily precipitation,
depending on the precipitation threshold considered.
Score differences between simulations with or without data assimilation are significant at 95% level
for most scores and thresholds considered, showing the positive and statistically robust impact of
the lightning data assimilation on the precipitation forecast.

**Key words:** total lightning data assimilation, forecast verification, convective storms, cloud
resolving model.

**1. Introduction**
The inclusion of the effects of deep convection in the initial conditions of Numerical Weather
Prediction (NWP) models is one of the most important problem to reduce the spin-up time and to
improve initial conditions (Stensrud and Fritsch, 1994; Alexander et al., 1999). In recent years,
several studies have shown the positive impact that lightning assimilation has on the weather
forecast, and especially on the precipitation forecast (Alexander et al. 1999; Chang et al., 2001;
Papadopulos et al., 2005; Mansell et al., 2007; Fierro et al., 2012; Giannaros et al., 2016).
Lightning data are a proxy for identifying the occurrence of deep convection, which relates to
convective precipitation (Goodman et al., 1988). In addition to their ability to locate precisely the
deep convection and heavy precipitation, lightning data have several advantages: availability in real
time with few gaps (reliability), compactness (a low band is required to transfer the data), long-
range detection of storms over the oceans and beyond the radars (Mansell et al., 2007).
Because of these properties, several techniques have been developed, in recent years, to assimilate
lightning data in NWP. In the first studies (Alexander et al. 1999; Chang et al., 2001), lightning
were used in conjunction with rainfall estimates from microwave data of polar orbiting satellites to
derive a relation between the cloud to ground flashes and rainfall. Then the rainfall estimated from
lightning was converted to latent heat nudging, that was assimilated in NWP (Jones and Macpherson,
1997). These experiments showed a positive impact of the lightning data assimilation on the 12-24
h weather forecast.



Papadopulos et al. (2005) nudged relative humidity profiles associated with deep convection and
the adjustment was proportional to the flash rate observed by the ZEUS network (Lagouvardos et
al., 2009).
A modification of the Kain-Fritsch (Kain and Fritsch, 1993) convective parameterization in
COAMPS (Coupled Ocean-Atmosphere Mesoscale Prediction System; Hodur, 1997) was
introduced by Mansell et al. (2007). They enabled lightning to control the cumulus parameterization
scheme activation. Recently, Giannaros et al. (2016) implemented a similar approach in the WRF
model, showing the positive and statistically robust impact of the lightning data assimilation on the
24h rainfall forecast for eight convective events over Greece.
Fierro et al. (2012) and Qie et al. (2014) show two lightning data assimilation schemes for the WRF
model intervening on the mixing ratios of the hydrometeors (water vapour in the case of Fierro et
al. (2012), and ice crystals, graupel and snow in Qie et al. (2014)). Both studies, which are made at
cloud resolving scales, show that lightning assimilation can improve the precipitation forecast.
Most of the studies cited above are based on a case study approach. However, Giannaros et al.
(2016) applied the methodology to eight convective cases occurred in Greece from 2010 to 2013.
Considering a larger number of cases allowed them to statistically test the improvement of the
precipitation forecast through lightning data assimilation. Moreover, their methodology is designed
to be realistic and usable in the operational forecast.
In a recent study, Federico et al. (2014) introduced a scheme to simulate lightning in the RAMS
model (Regional Atmospheric Modeling System). Because the lightning distribution is well
correlated to areas of deep convection, they concluded that lightning simulation can be a useful tool
to evaluate the reliability of the NWP forecast in real time. In their study, however, lightning
observations were used as a diagnostic tool.
In this paper, a total lightning data assimilation algorithm is used in the RAMS model. The
assimilation scheme is similar to that of Fierro et al. (2012), with few modifications to account for
different spatial and temporal resolutions of the two studies and for the different model suites. In
addition, the methodology presented in this paper is designed to be used in real time NWP. This
paper considers the short-term forecast (3h), even if the results for daily precipitation, accumulated
from the 3h precipitation forecast, are also shown for completeness and for comparison with other
studies.
To evaluate statistically the impact of the lightning data assimilation on the precipitation forecast,
we consider twenty convective cases occurred in fall 2012 over Italy, which is the target of this
study. Most of these events occurred during the HyMeX SOP1 (Hydrological cycle in the


Mediterranean Experiment – First Special Observing Period), which was held from 5 September
2012 to 6 November 2012.
HyMeX (Drobinski et al., 2014; Ducroq et al., 2014) is an international experimental program that
aims to advance scientific knowledge of water cycle variability in the Mediterranean basin. This
goal is pursued through monitoring, analysis and modeling of the regional hydrological cycle in a
seamless approach. In HyMeX special emphasis is given to the topics of the occurrence of heavy
precipitation and floods, and their societal impacts, which were the subjects of the SOP1. One of the
products of the HyMeX-SOP1 is a database of hourly precipitation available for 2944 raingauges
over Italy belonging to the Italian DPC (Department of Civil Protection; Davolio et al., 2015). This
database extends behind the period of the HyMeX-SOP1 and contains all the events considered in
this paper.
The paper is organized as follows: Section 2 shows the RAMS configuration, the methodology used
to assimilate total lightning data, and the strategy used in the simulations. Section 3 gives the
results: first a case study of deep convection occurred over Italy during HyMeX-SOP1 is considered
to show how the lightning data assimilation works (Section 3.1); then the scores for the twenty
cases are shown in Section 3.2, which also shows the statistical robustness of the difference
between the precipitation forecasts of the simulations with or without total lightning data
assimilation. The discussion and conclusions are given in Section 4.

**2. Methodology**
*2.1 The RAMS model configuration*
The RAMS model is used in this study. This section is a brief description of the model setup, while
details on the model are given in Cotton et al. (2003).
We use two one-way nested domains at 10 km (R10) and 4 km (R4) horizontal resolutions,
respectively (Table 1, see Figure 2a for the domain at 10 km horizontal resolution and Figure 3a for
the domain at 4 km horizontal resolution). The model is configured with thirty-six terrain following
vertical levels for both domains. The model top is at 22400 m. The distance of the levels is
gradually increased from 50 m to 1200 m. Below 1000 m the spacing between levels is less than
200 m, above 5000 m the distance between levels is 1200 m.
The Land Ecosystem-Atmosphere Feedback model (LEAF) is used to calculate the exchange
between soil, vegetation, and atmosphere (Walko et al., 2000). LEAF uses a patch representation of





surface features (vegetation, soil, lakes and oceans, and snow cover) and includes several terms
describing their interactions as well as their exchanges with the atmosphere.
Explicitly resolved precipitation is computed by the WRF (Weather Research and Forecasting
System) – single-moment-microphysics class 6 (WSM6) scheme (Hong et al., 2006). This scheme
was recently implemented in RAMS (Federico, 2016) and showed the best performance among the
microphysical schemes available in the model for a forecast period spanning 50 days of the
HyMeX-SOP1 at 4 km horizontal resolution. The WSM6 scheme accounts for the following water
variables: vapour, cloud water, cloud ice, rain, snow and graupel. The best configuration of
Federico (2016) is used in this paper and is hereafter referred to as control (CNTRL).
Sub-grid-scale effect of clouds is parameterized following Molinari and Corsetti (1985). They
proposed a form of the Kuo scheme (Kuo, 1974) accounting for updrafts and downdrafts. The
convective scheme is applied to the 10 km grid only.
The unresolved transport is parametrized by the K-theory following Smagorinsky (1963), which
relates the mixing coefficients to the fluid strain rate and includes corrections for the influence of
the Brunt-Vaisala frequency and the Richardson number (Pielke, 2002).
The Chen and Cotton (Chen and Cotton, 1983) scheme is used to compute short and long-wave
radiation. The scheme accounts for condensate in the atmosphere, but not for the specific
hydrometeor type.
The initial and dynamic boundary conditions are introduced in section 2.3.

*2.2 Lightning data and assimilation procedure*
Lightning data used in this paper are those observed by LINET (LIghtning detection NETwork;
Betz et al., 2009), which is a European lightning location network for high-precision detection of
total lightning, ground strokes (exchanging charges between the cloud and the ground - CG cloud-
to-ground) and cloud lightning (not making ground contact - IC intra cloud), with utilization of
VLF/LF techniques (in range between 1 and 200 KHz).
The network has more than 550 sensors in several countries worldwide, with very good coverage
over central Europe and central and western Mediterranean (from 10° W to 35° E in longitude and
from 30° N to 65° N in latitude). The lightning three-dimensional location is detected using the time
of arrival (TOA) difference triangulation technique (Betz et al., 2009). The lightning strokes are
detected with high precision (150 m for an average distance between sensors of 200 km) in both





horizontal and vertical directions. The LINET "strokes" are grouped into "flashes" before
assimilation in the model. In particular, all events recorded by LINET that occur within 1 s and in
an area with a radius of 10 km are binned into a single flash (Federico et al., 2014).
Observed flashes are mapped onto the RAMS grid for assimilation in space and time. In particular,
the assimilation procedure computes the number of flashes occurring in each RAMS grid cell in the
past five minutes ($X$). Then the water vapour mixing ratio is computed as:
$$q_v = Aq_s + B*q_s*tanh(CX)*(1-tanh(DQ_g^{\alpha})) \tag{1}$$
Where $A$=0.86, $B$=0.15, $C$=0.30 $D$=0.25, $\alpha$=2.2, $q_s$ is the saturation mixing ratio at the model
atmospheric temperature, and $Q_g$ is the graupel mixing ratio (g kg$^{-1}$). The water vapour mixing ratio
derived from Eqn. (1) is similar to Fierro et al. (2012). There are two changes: first the $C$ coefficient
is larger in this study (in Fierro et al. (2012), C=0.01), which partially accounts for the different
horizontal resolutions of the remapped observed flashes (9 km in Fierro et al., (2012); 4 km in our
case, corresponding to the RAMS inner grid horizontal resolution) and for the different grouping
time interval (10 minutes in Fierro et al. (2012), and 5 minutes here). Second, the coefficient $A$ ($B$)
is larger (smaller) in this study compared to Fierro et al. (2012; $A$=0.81 and $B$=0.20) because we
find a better performance with this set-up. The set-up of Eqn. (1) was found by trials and errors for
two case studies (15 and 27 October 2012) by considering two opposite needs: to increase the
precipitation hits and to reduce (or not increase considerably) the false alarms. It is noted that Fierro
et al. (2012) found little sensitivity of the results by varying $A$ by 5%.
The water vapour derived from Eqn. (1) is substituted to the simulated value at a grid point where
electric activity is observed and RH is below 86%. By this choice we only add water vapour to the
simulated field, leaving it unchanged if the simulated water vapour is larger than that of Eqn. (1).
Moreover, the water vapour is substituted only in the charging zone (from 0 to -25 °C), which is the
mixed-phase graupel-rich zone associated with electrification and lightning activity (MacGorman
and Rust, 1998). The increase of $q_v$, Eqn. (1), is inversely proportional to the simulated graupel
mixing ratio. When $Q_g$ is 3 g/kg the application of Eqn. (1) is ineffective (see Figure 7 of Fierro et
al. (2012) for the dependency Eqn. (1) on the graupel mixing ratio). For a given value of $Q_g$
between 0 and 3 g/kg, the water vapour of Eqn. (1) increases as a function of the gridded flash rate
$X$.


*2.3 Simulation strategy and verification*
Twenty case studies occurred in fall 2012 were selected. The events are reported in Table 2 and
were all characterized by widespread convection, lightning activity, and moderate-heavy
precipitation over Italy. The events of Table 2 comprise eight of the nine IOP (Intense Observing
Period) declared in Italy (see Table 5 of Ferretti et al. (2014) for the complete list of the IOP) during
HyMeX-SOP1 and few other cases of November 2012.
A 36 h forecast at 10 km horizontal resolution is performed for each case (R10). The initial and
boundary conditions (BC) for this run are given by the 12 UTC assimilation/forecast cycle of the
ECMWF (European Centre for Medium Weather range Forecast). Initial and BC are available at
0.25° horizontal resolution. The R10 forecast starts at 12 UTC of the day before the day of interest
(actual day, Table 2) and the first 12 hours, which also account for the spin-up time, are discarded
from the evaluation. The R10 forecast is made to give the initial and BC to the 4 km horizontal
resolution forecast (R4), avoiding the abrupt change of resolution from the ECMWF initial
conditions and BC (0.25°) to the R4 horizontal resolution.
Starting from R10 as initial and BC, three kind of simulations, all using the R4 configuration, are
performed for each event: a) CNTRL: this simulation is performed by nesting R4 in R10 using a
one-way nest and without doing lightning data assimilation. Each CNTRL simulation starts at 18
UTC of the day before the actual day and the first six hours, which account for the spin-up time, are
discarded from the evaluation; b) F3HA6: these simulations consist of eight runs of 9 h duration.
During the first 6 h, lightning data are assimilated following the procedure described in the previous
section.  Then, a short term 3 h forecast is made. Eight F3HA6 simulations are needed to span the
forecast of a whole day (Figure 1). The first simulation starts at 18 UTC of the day before the actual
day, using as initial and boundary conditions the R10 forecast, and gives the forecast for the hours
00-03 UTC of the actual day. The second F3HA6 simulation starts at 21 UTC of the day before the
actual day using as initial conditions the previous R4 forecast and as BC the R10 forecast.
Lightning are assimilated from 21 UTC of the day before to 03 UTC of the actual day, while the
forecast is valid for 03-06 UTC of the actual day. The F3HA6 forecasts from three to eight proceed
as the second but shifted every time three hours ahead. It is noted the switch of the initial conditions
between the first and second F3HA6 simulations from R10 to R4. This is done to maximise the
impact of lightning data assimilation on the F3HA6 run, since the initial conditions provided by R4
are produced by a simulation using lightning data, while in R10 lightning data are not used; c)
ASSIM: this simulation is performed by nesting R4 in R10 using a one-way nest and doing
lightning data assimilation for the whole run. Each ASSIM simulation starts at 18 UTC of the day

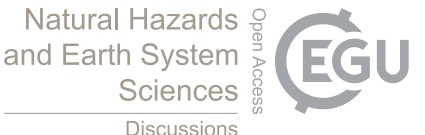

before the actual day and the first six hours of forecast are considered as spin-up time and are
discarded from the evaluation. The ASSIM simulation continuously assimilates lightning data and,
because it represents better the convection during the events compared to CNTRL and F3HA6, has
the best performance (Section 3.2). The ASSIM configuration can be useful when analysing the
events but cannot be used for the forecast because it needs real-time lightning data as the integration
time advances.
It is noted that the configuration F3HA6 was chosen because it can be applied in the operational
context. The simulation R10 takes less than one hour to complete the 36 h forecast on a 64 core
state of the art cluster. Each simulation F3HA6 takes 20-25 minutes using a 64 cores state of the art
cluster, which makes the forecast available for operational purposes.  Continuous advancing of
computing power will give the possibility to apply the methodology at finer horizontal resolutions
for extended areas, as that considered in this paper, as well as to reach the kilometric scale for
limited areas.
Even if the main focus of this paper is on the short-term (3 h) forecast, the daily precipitation
accumulated from the 3h forecasts is also considered for comparison with other studies available in
the literature. For F3HA6 the daily precipitation is given by adding the eight 3 h forecasts available
for the actual day (Figure 1).
One of the products of the HyMeX initiative is a database of hourly precipitation available from the
network of the DPC of Italy, which consists of 2944 raingauges all over Italy. The dataset is
available                                                                                          at
http://mistrals.sedoo.fr/?editDatsId=1282&datsId=1282&project_name=MISTR&q=DPC and it is
used to derive 3 h and daily rainfall, which are then used to verify the model.
For the verification of the QPF, the model output at the closest grid point of a raingauge is
considered.  When two or more raingauge stations fall in the same model grid-cell the average
precipitation recorded by these stations is considered.
Statistical verification is performed by 2x2 contingency tables for different precipitation thresholds.
For the 3 h rainfall comparison the thresholds are: 0.2, 1.0, 3.0, 5.0, 7.5, 10.0, 15.0, 20 mm/3h. For
daily precipitation the thresholds are: 1, 5, 10, 20, 30, 40, 60 mm/day, being 60 mm/day (7.5
mm/3h) considered as the threshold for severe precipitation events in the Mediterranean Basin
(Jansà et al., 2014). From the hits ($a$), false alarms ($b$), misses ($c$), and correct no forecasts ($d$) of the
contingency tables, the probability of detection (POD; range [0, 1], where 1 is the perfect score, i.e.
when no misses and false alarms occur), the False Alarm Ratio (FAR; range [0, 1], where 0 is the
perfect score), the Bias (range [0, $+\infty$), where 1 is the perfect score) and the equitable threat score





(ETS; range [-1/3,1], where 1 is the perfect score and 0 is a useless forecast) are computed (Wilks,

256   2006):

$$
\begin{aligned}
POD &= \frac{a}{a+c} \\
FAR &= \frac{b}{a+b} \\
Bias &= \frac{a+b}{a+c} \\
ETS &= \frac{a-a_r}{a+b+c-a_r}; \quad and \quad a_r = \frac{(a+b)(a+c)}{a+b+c+d}
\end{aligned}
\tag{1}
$$


where $a_r$ is the probability to have a correct forecast by chance (Wilks, 2006).
The Bias tells us the fraction of rain forecast events with respect to the rain observed events. The
POD gives the fraction of the observed rain events that were correctly forecast. The FAR gives the
faction of rain forecast events that didn't occur. The ETS measures the fraction of observed and/or
forecast rain events that were correctly predicted, adjusted for hits associated to a random forecast,
where the forecast occurrence/non-occurrence is independent of observation/non observation.
In order to have a measure of the difference between the CNTRL and F3HA6 forecast a hypothesis
test to verify that the score difference between the two competing models is significant at a
predefined significance level (90%, α=0.1; or 95%, α=0.05) is made. The test was originally
proposed by Hamill (1999) and is based on resampling.  The null hypothesis of the resampling test
is that the difference of the scores between the competitor forecasts is zero. The score is computed
from the sum of the contingency tables available (8 multiplied the number of cases. i.e. 20*8=160
for the 3h precipitation forecast; and 20 for the daily precipitation forecast) to minimize the
sensitivity of the test to small changes of the contingency table elements.
A random sampling of the contingency table elements was performed 10.000 times using the
bootstrapping technique, as detailed in Accadia et al. (2003) and Federico et al. (2003). Each time
the scores are computed from the sum of the elements of the resampled contingency tables to make
the null distribution of the difference between the scores of the competitor forecasts.
Then we compute the $t_L$ and $t_U$ that represent the α/2 and (1- α)/2 percentile of the null distribution
$(S_1{}^*-S_2{}^*)$ where $S_1{}^*$ and $S_2{}^*$ are the generic scores of the resampled distributions. The null hypothesis
that the score difference between the two competitor forecasts is zero is rejected at the level 90 %
(α=0.1) or 95% (α=0.05) if:

280                      $(S_1 - S_2) < t_L$   or   $(S_1 - S_2) > t_U$





where $S_1$ and $S_2$ are the generic scores of the actual distributions (not resampled).

**3. Results**

*3.1 The 27 October 2012 case study*
The event studied in this section is taken from the HyMeX SOP1 campaign, which was focused on
heavy precipitation and its societal impact (Ducroq et al., 2014; Ferretti et al., 2014). Nine of the
twenty IOPs (Intense Observing Period) considered in SOP1 occurred in Italy.
During SOP1, several through extended toward the Mediterranean Basin or entered in the Basin as
deep through. Few of them developed a cut-off, while most of them generated a low pressure
pattern in Northern Italy, which usually moved along the Italian peninsula. The 27 October 2012
case study, also referred as IOP16a, belongs to the latter class of events, but it eventually evolved in
a cut-off on 28-29 October (IOP16c). This event, characterized by widespread convection and
intense lightning activity, caused huge precipitation all along the peninsula and also peak values of
water level on the Venice Lagoon, where the sea level exceeded twice the warning level of 120 cm
(Casaioli et al., 2013; Mariani et al., 2014).
Figure 2 shows the synoptic situation at 12 UTC on 27 October 2012. At 500 hPa, Figure 2a, a
trough extends from NE Europe toward the Western Mediterranean. The interaction between the
trough and the Alps generated a mesolow over northern Italy, as shown by the 990 hPa contour in
Figure 2b, that caused a cyclonic circulation over most of the peninsula.
In these synoptic conditions, winds over the Tyrrhenian Sea are from W and SW and bring humid
and unstable air over the mainland of Italy. The interaction between the unstable air and the
orography of Italy reinforced the convection, which, however, was already occurring over the sea as
shown by the intense electric activity over the Tyrrhenian Sea (see below).
Figure 3a shows the lightning distribution observed by LINET on 27 October 2012. From Figure 3a
it is well evident the convection over the Tyrrhenian Sea, which is also active over the land because
of the interaction between the humid and unstable air masses from the sea and the orography of
Italy.
The daily precipitation (Figure 3b), which is unavailable for a wide area of Central-Northern Italy
shows the widespread convection over the Apennines, with several stations reporting more than 90
mm/day. Note also the abundant precipitation over Sardinia and over the North-East of Italy. It is
important to note that the rainfall of Figure 3b is computed by summing the 1h precipitation



registered by the raingauges. If one of the 1h observations is unavailable, the raingauge does not
appear in Figure 3b. So, when verifying the precipitation for shorter time scales, different
raingauges could appear compared to those of Figure 3b.
Figures 4a and 4b show the daily precipitation forecast of the CNTRL run and the daily
accumulated precipitation of the F3HA6 run. Figures 4a and 4b shows a high precipitation amount
over the Apennines (> 90 mm/day) all along the peninsula, in agreement with observations.
However, the precipitation area for the largest threshold is overestimated by both CNTRL and
F3HA6. This is apparent by comparing the area of the 90 mm/day threshold in Figures 4a-4b with
the comparatively few raingauges reporting this precipitation amount. As it will be shown in the
next section, this is a general behaviour of the RAMS model with the set-up used in this paper.
Other features shown by Figures 4a and 4b are: a very heavy precipitation spell in NE Italy,  whose
area is overestimated by CNTRL and F3HA6; a high precipitation spell over the Liguria-Tuscany
area, which is only partially revealed by observations due to the lack of data; a moderate
precipitation over Sardinia, which is underestimated by the CNTRL forecast both for the
precipitation area and amount.
Even if CNTRL and F3HA6 share several precipitation features in common, there are important
differences between Figures 4a and 4b. The convection over the sea is underestimated by CNTRL.
Even if we cannot prove it by the precipitation amount, the intense electrical activity over the
Central Mediterranean Sea, and especially over the Tyrrhenian Sea, shows that the convective
activity over the sea is underestimated by CNTRL.
The convection over the sea is simulated by F3HA6 thanks to the lightning data assimilation. When
the convection is advected over the land it increases the precipitation. This is clearly shown by the
precipitation over Sardinia, which increases both in areal coverage and rainfall amount for F3HA6
compared to CNTRL.
Other differences between the precipitation field of CNTRL and F3HA6 can be discussed more
easily by the difference of the precipitation fields. Figure 4c shows the precipitation difference
between CNTRL and F3HA6 in this order, so that positive values show larger precipitation for
CNTRL, while negative values show larger precipitation for F3HA6.
From Figure 4c it is apparent that the precipitation of F3HA6 increases over large areas of the
domain, especially over the Tyrrhenian Sea. The rainfall over Sardinia increases up to 40 mm/day,
showing the important impact of the lightning assimilation on the forecast. However, the largest
differences are found along the Apennines with values up to 80 mm/day.





In general, the lightning assimilation increases the precipitation, nonetheless Figure 4c shows also
areas where the precipitation of F3HA6 decreases compared to CNTRL, because of the different
evolution of the storm in the two simulations. This is especially evident over the Adriatic coast of
the Balkans where positive-negative patterns alternate every few tens of kilometres. We will discuss
further this point later on in this section.
Up to now, we considered the impact of the lightning assimilation on the daily precipitation, i.e.
when the rainfall of the eight F3HA6 forecasts in a day are added, however the main focus of this
paper is on the short-term precipitation forecast. To consider this point, Figure 5a shows the
observed precipitation accumulated between 06 and 09 UTC, and the corresponding precipitation
for the CNTRL (Figure 5b) and F3HA6 (Figure 5c).
Figure 5a shows a considerable precipitation spells (about 40 mm/3h) over NE Italy, in some spots
over the Apennines all along Italy, and, somewhat smaller, over Sardinia.
Comparing Figure 5b with Figure 5a it is apparent that the CNTRL forecast is able to catch several
features of the precipitation field, as the local spots of heavy rain over the Apennines or the rain
spell over NE Italy, the main error being the scarce precipitation simulated over Sardinia. This issue
is in part solved by the F3HA6 forecast (Figure 5c), which shows larger precipitation compared to
CNRTL over Sardinia.
To better focus on the improvement given by the lightning data assimilation on the short term QPF
we consider the precipitation hits, i.e. the correct forecasts, of the contingency tables. Figure 6a
shows the difference between the hits of the F3HA6 and CNTRL (in this order) for the 1 mm/3h (8
mm/day) threshold. In Figure 6a, the +1 (red asterisk) shows a station where the CNTRL forecast
did not predict a precipitation equal or larger than the threshold, while the F3HA6 correctly
predicted a rainfall equal or larger than the threshold at the raingauge. The -1 value (blue asterisk)
shows the opposite behaviour.  In Figure 6a there are fifty-two new correctly predicted events for
F3HA6. They are located in the Apennines and, mostly, over Sardinia, where CNTRL missed the
forecast (Figures 4a-4b). There are also two stations where the lightning assimilation worsens the
forecast, because of the different evolutions of the storms in CNTRL and F3HA6, nevertheless the
benefits of the lightning data assimilation on the short term QPF are apparent for the 1 mm/3h
threshold.
Figure 6b shows the difference between the hits of F3HA6 and CNTRL for the 10 mm/3h (80
mm/day) threshold, which is more interesting when considering moderate-high rainfall amounts.
For this threshold, the lightning data assimilation improves the forecast because twelve new events
are correctly predicted by F3HA6 along the Apennines and over Sardinia.





It is important to note the precision of the correction to the precipitation field given by the lightning
data assimilation. The positive-negative pattern of the difference between the precipitation fields of
CNTRL and F3HA6 (shown for the daily precipitation, Figure 4c, with amplitudes of tens of
kilometres in the Central Apennines) is found, with lower amplitude, also for the 3h forecast (not
shown). The F3HA6 forecast gave the correct prediction of several new stations for both 1 mm/3h
(fifty-two raingauges) and 10 mm/3h (twelve raingauges) thresholds, while losing only two stations
correctly predicted by CNTRL for the 1 mm/3h threshold. This shows that the precipitation is added
where necessary, but also that it is subtracted where it did not occur, i.e. only two correct forecasts
are lost by the lightning data assimilation. It is worth noting that the stations correctly forecast by
both CNTRL and F3HA6 for a given precipitation threshold do not appear in Figures 6a and 6b.
This occurs, for example, for the raingauges in NE Italy.
This section showed how the data assimilation technique of this study works and how it is able to
add new correct forecasts (hits) to CNTRL for a case study. In the following section, scores based
on contingency tables are presented for a total of twenty case studies in order to quantify, in a
statistically robust way, the benefits of the total lightning data assimilation on the short-term QPF.

*3.2 Statistical scores*
In this section we discuss the statistical scores of the F3HA6 forecast in comparison to CNTRL.
The results of the ASSIM run are also presented as the benchmark for lightning data assimilation.
First we discuss the results for the daily precipitation accumulated starting from 3h rainfall
forecasts.
Figure 7a shows that the Bias increases from 0.8-1.0 (1 mm/day threshold, depending on the type of
simulation) to 2.3-2.6 (60 mm/day threshold), showing a considerable overestimation of the forecast
area for the larger thresholds (> 40 mm/day). The lightning data assimilation improves the Bias up
to 10 mm/day (both F3HA6 and ASSIM), while performance is worsened by data assimilation for
larger thresholds. As expected the ASSIM shows the largest Bias, followed by F3HA6 and CNTRL.
This is caused by the addition of water vapour by the data assimilation, which is larger for ASSIM
(assimilation performed continuously) compared to F3HA6 (assimilation is not performed in the
forecast phase). The statistical test to assess the bias difference between CNTRL and F3HA6 shows
that the two scores are different at 95% significance level for all thresholds, showing the significant
impact of the lightning data assimilation on the precipitation forecast.
The overestimation of the precipitation area for higher thresholds is well evident, as discussed in the
previous section, in Figures 4a-4b over the Apennines for the 90 mm/day threshold (the ASSIM



simulation, not shown, does not differ substantially from F3HA6). Comparing the result of the Bias with the same result of Federico (2016), where the same configuration of the RAMS model of CNTRL was used, we note a considerable increase of the Bias in this work. This difference is caused by the fact that Federico (2016) considered 50 consecutive days of the HyMeX-SOP1, i.e. with heavy, moderate and small precipitation, while this study considers only cases with deep and widespread convection. The RAMS with WSM6 scheme shows the tendency to overestimate the Bias for increasing precipitation (Federico, 2016; see also Liu et al., 2011 for a general comparison of the WSM6 microphysical scheme and other microphysical schemes available in the Weather Research and Forecast (WRF) model), and this tendency is amplified for the heavy precipitation events considered in this work.

Figure 7b shows the ETS score. For CNTRL it decreases from 0.35 (1 mm/day) to 0.17 (60 mm/day). The ETS increases for F3HA6, especially for thresholds lower than 30 mm/day, showing the positive impact of the lightning assimilation on the precipitation forecast. The difference of the ETS for F3HA6 and CNTRL is statistically significant at 90% level for the 30 mm/day threshold, at 95% level for lower precipitation, and not significant for larger precipitation. The ASSIM simulations show a further increase of the ETS compared to F3HA6 because of their ability to better represent the convection during the simulation through lightning data assimilation.

The POD (Figure 7c) for CNTRL decreases from 0.70 (1 mm/day) to 0.52 (60 mm/day), i.e. half of the potentially dangerous events are correctly predicted. It is also noted the rather stable value of the POD (0.6) between the 10 and 40 mm/day thresholds. The POD increases for F3HA6. The lowest increment is attained for 60 mm/day (0.04, i.e. 4% more potentially dangerous events are correctly forecast compared to CNTRL), the largest for the 1 mm/day (6.5%). Differences between the POD of CNTRL and F3HA6 are significant at 95% level for all thresholds showing the robust improvement of the performance for this score using lightning data assimilation. Notably, the ASSIM run increases the POD of 8-10%, depending on the threshold.

The FAR for CNTRL (Figure 7d) increases from less than 0.2 (1 mm/day threshold; i.e. less than 20% of the forecasts are false alarms) to 0.8 (60 mm/day threshold; i.e. 80% of the forecasts are false alarms). The lightning assimilation improves the performance for the FAR but differences are statistically significant for 1 mm/day (90% level), 5 and 10 mm/day (95% level). The inspection of the contingency tables shows that the improvement of the FAR for those thresholds is attained by a larger number of hits but there is also an increase of the false alarms.

Figure 8a shows the Bias for the 3h precipitation forecast. The Bias for CNTRL increases from about 1 (0.2 mm/3h threshold) to 2.5 (20 mm/3h threshold). The Bias increases for F3HA6 and





ASSIM compared to CNTRL and the Bias differences between CNTRL and F3HA6 are significant at 95% level for all thresholds.

The ETS score (Figure 8b) for CNTRL shows a decrease from 0.33 (0.2 mm/3h threshold) to 0.13 (20 mm/3h threshold). The ETS is larger for F3HA6 compared to CNTRL and the differences of the scores are significant at 95% level for all thresholds. The ETS of ASSIM, as expected, is larger than that of F3HA6. It is also noted that, while the ETS is positive for all thresholds, showing the usefulness of the forecast, the ETS value is rather low for the 20 mm/3h threshold. This is mainly caused by the large number of false alarms for this threshold.

Figure 8c shows the POD for the 3h forecast. The value for CNTRL decreases from 0.63 (0.2 mm/3h) to 0.43 (20 mm/3h). The POD increases for F3HA6, notably for thresholds up to 7.5 mm/3h (>5%), while the improvement is smaller (3%-4%) for larger thresholds. The score difference between F3HA6 and CNTRL is statistically significant at 95% level for all thresholds.

Figure 8d shows the FAR for the 3h forecast. The FAR increases from 0.3 to 0.83 for the CNTRL forecast, showing again the tendency of the false alarms to increase with increasing precipitation thresholds. The FAR for F3HA6 decreases (1-3% depending on the threshold) and the differences of the FAR for CNTRL and F3HA6 are statistically significant at 95% level up to the 7.5 mm/3h threshold and at 90% significance level for 10 mm/3h and 20 mm/3h threshold. As for the daily precipitation forecast, the FAR improvement for F3HA6 is the result of the increase of the hits but it is also associated to an increase of the false alarms.

## 4. Discussion and conclusions

This study shows the application of a total lightning data assimilation technique, developed by Fierro et al. (2012), to the RAMS model with WSM6 microphysics scheme (Federico, 2016). The technique adds water vapour to grid columns where flashes are observed, and the water vapour added at constant temperature depends on the flash rate and on the graupel mixing ratio. Water vapour is added to the model when suitable, while the water vapour is unchanged when the model predicts a value larger than that of the data assimilation algorithm. This paper shows a realistic implementation of the assimilation/forecast procedure that can be adopted in operational weather forecast.

The results of this paper show that the methodology is effective at improving the short-term (3h) precipitation forecast. More in detail, the analysis of the 27 October shows that the total lightning data assimilation is able to trigger the convection over the sea and, when the convection is advected



over the land, it improves the short-term precipitation forecast. This effect is apparent over Sardinia
for the case study.  The humid marine air masses interact with the local orography causing or
reinforcing the convection. Also, the lightning data assimilation improves the rainfall forecast
adding precipitation where it is observed and increasing the hits of the short-term forecast.
The analysis of the scores for the 3h precipitation forecast, computed for twenty cases characterized
by intense lightning activity and widespread convection, confirms the improvement of the
precipitation forecast using lightning data assimilation. The ETS and POD increase for all
thresholds considered for F3HA6 compared to CNTRL and the difference between the scores of the
competitor forecasts is significant at 95% level for all thresholds. The FAR is also improved and the
difference between the scores of F3HA6 and CNTRL is statistically significant for all thresholds
with the exception of the 15 mm/3h. The FAR improvement of F3HA6 is caused by the increase of
the hits, but it is also associated to a larger number of false alarms.
The Bias is the only score that worsens with lightning data assimilation. The Bias of the RAMS
model with the WSM6 microphysics scheme is larger than one for most thresholds for the case
studies of this paper. Because the lightning data assimilation adds water vapour to the model, the
tendency to overestimate the precipitation area, especially for the larger thresholds, is worsened by
the lightning data assimilation.
In addition to the 3h forecast, the scores and precipitation field are analysed for the daily
precipitation for completeness and for comparison with other studies. Recently, Giannaros et al.
(2016) presented the WRF-LTGDA, a lightning data assimilation technique implemented in WRF.
They presented the results for eight cases in Greece. Their assimilation strategy focuses on the daily
rainfall prediction (tomorrow daily precipitation). Their analysis (see their Figure 3, note also that
the maximum precipitation threshold is 20 mm/day in their study) shows that the POD increases
when lightning data assimilation is compared to CNTRL, and the increase of the POD is up to 5%.
Moreover, for some thresholds, the lightning assimilation lowers the POD because of the different
patterns followed by the storms in the simulations with or without lightning data assimilation.
Our results show that the POD improves for all precipitation thresholds when lightning data
assimilation is used and the percentage of improvement is slightly better than that reported in
Giannaros et al. (2016) for the lower thresholds (below 10 mm/day). Even if we cannot give a
definitive answer to this issue, because of the many important differences between this study and
that of Giannaros et al. (2016), the lightning data assimilation technique has a role. In our case,
lightning data are assimilated also for the actual day (6h assimilation before the forecast start time
followed by 3h forecast, Figure 1), while in Giannaros et al. (2016) the assimilation is done only for



the day before the actual day (6h assimilation followed by 24 h forecast). So, our technique should
improve the correct location of the convection during the actual day compared to their approach, as
shown by the slightly larger improvement, i.e. the difference between the POD of the simulations
with or without lightning data assimilation.
However, other differences play a role: first the two studies refer to different regions and to
different events. In our case the extension of the region, the number of the events, and the number
of verifying stations are larger. Moreover, two different model suites are used (WRF and RAMS).
These differences are clearly seen in the score values. The POD of Giannaros et al. (2016), is larger
than that of this study, especially for thresholds lower than 20 mm/day. Another important
difference arises from the different convective nature of the storms considered in the two works. In
Giannaros et al. (2016) it is clearly shown the dependence of the performance of the precipitation
forecast on the type of event, i.e. widespread or localized convection, and, because the events
considered in the two studies are different, the comparison between the two works can be only
qualitative. Nevertheless, both studies show that the lightning data assimilation improves the
precipitation forecast robustly, and can be used in the operational context.
While the results of this study are encouraging, there are a number of issues that need further
investigation. The water vapour is added to the grid column where the lightning is observed.
However, the lightning is often the result of a process involving larger scales than the horizontal
grid spacing considered in this paper (4 km). A spatial extension of the influence of the lightning
perturbation on the water vapour field should be explored. For this approach the applications of the
methods involving the model error matrix are foreseeable and will be investigated in future studies.
The problem of the spatial extension of the water vapour perturbation caused by lightning to the
model was considered in Fierro et al. (2012) by remapping the flashes onto a coarser horizontal
resolution grid (9 km), while no similar approach is done in this study.
A problem arising with the RAMS model using the WSM6 microphysics scheme is the
overestimation of the precipitation area for large rainfall thresholds. This tendency was already
noted in Federico (2016), and it is amplified for the cases of widespread convection considered in
this study. The high number of false alarms decreases the ETS score for high precipitation, reducing
the applicability of the method for the largest thresholds (> 100 mm/day). The application of
different microphysical schemes could mitigate this issue. Finally, higher horizontal resolutions are
also needed to better resolve the local orography and the interaction of the air masses with the
orography.







**Acknowledgments**

This work is a contribution to the HyMeX program. The author acknowledges Meteo-France and the HyMeX program for supplying the data, sponsored by Grants MISTRALS/HyMeX and ANR-11-BS56-0005 IODA-MED project. The ECMWF and Aeronautica Militare – CNMCA are acknowledged for the access to the MARS database. LINET data were provided by Nowcast GmbH (https://www.nowcast.de/) within a scientific agreement between Prof. H.-D. Betz and the Satellite Meteorological Group of CNR-ISAC in Rome.

551

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





**Tables**

Table 1: RAMS grid-setting for R10 and R4. NNXP, NNYP and NNYZ are the number of grid points in the west-east, north-south, and vertical directions. Lx(km), Ly(km), Lz(m) are the domain extension in the west-east, north-south, and vertical directions. DX(km) and DY(km) are the horizontal grid resolutions in the west-east and north-south directions. CENTLON and CENTLAT are the geographical coordinates of the grid centres.

|  | **R10** | **R4** |
| --- | --- | --- |
| NNXP | 301 | 401 |
| NNYP | 301 | 401 |
| NNZP | 36 | 36 |
| Lx | 3000 km | 1600 km |
| Ly | 3000 km | 1600 km |
| Lz | ~22400 m | ~22400 m |
| DX | 10 km | 4 km |
| DY | 10 km | 4 km |
| CENTLAT (°) | 43.0 N | 43.0 N |
| CENTLON (°) | 12.5 E | 12.5 E |

Table 2: The twenty case studies.

| Month | Days |
| --- | --- |
| September 2012 | 12,13,14,24,26,30 |
| October 2012 | 12,13,15,26,27,28,29,31 |
| November 2012 | 4,5,11,20,21,28 |




**Figures**

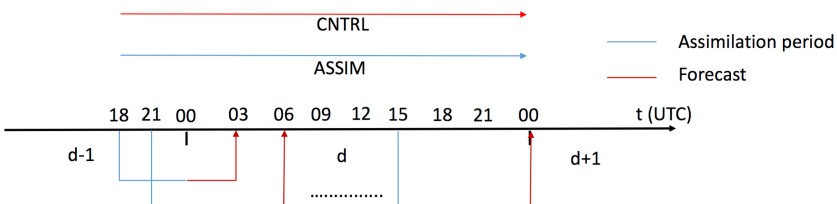



Figure 1: Synopsis of the simulations F3HA6 (below the timeline). The blue line is the assimilation
stage, while the red line is the forecast stage; d, d+1 and d-1 are the actual day, the day after and the
day before the actual day, respectively. In the upper part of the figure the CNTRL and ASSIM
simulations are shown.

**a)**

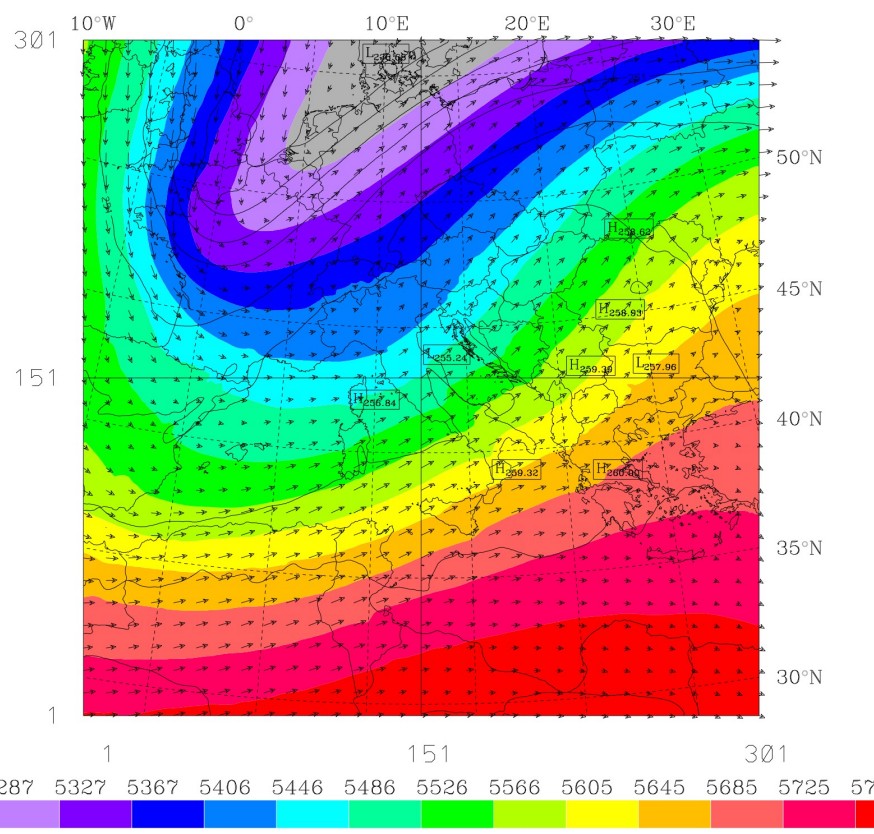







**b)**


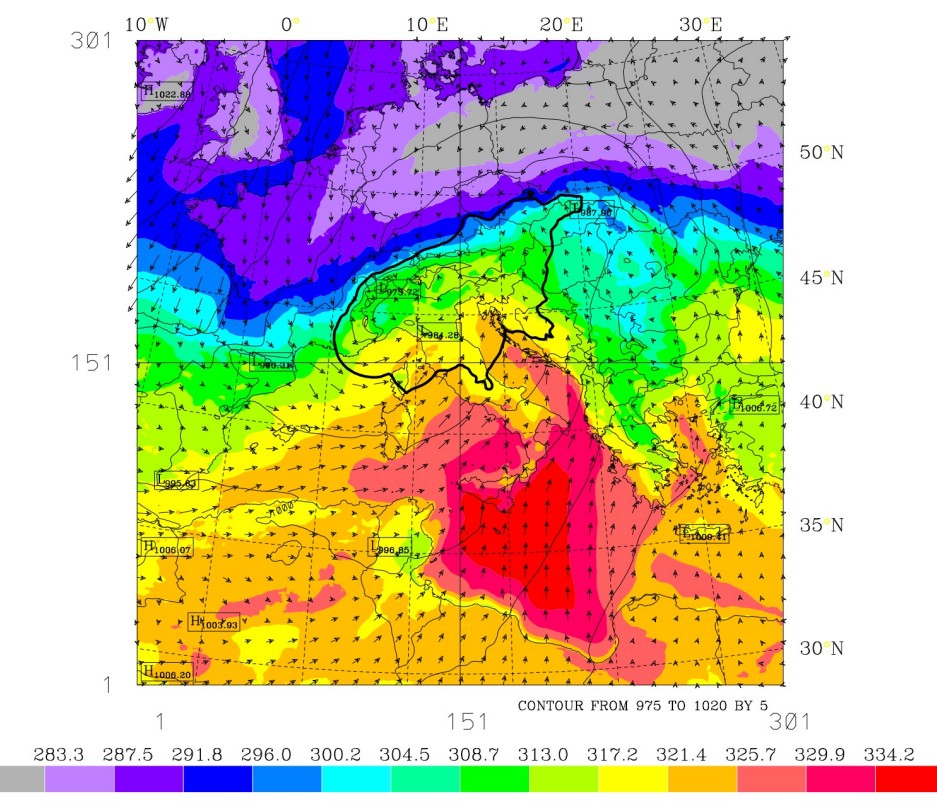


Figure 2: Synoptic situation at 12 UTC on 27 October 2012; a) 500 hPa: temperature (black contours from 236 K to 269 K every 3 K), geopotential height (filled contours, values shown by the colour bar at the bottom) and wind vectors (maximum wind value 41 m/s); b) surface: Sea level pressure (contour from 975 to 1020 hPa every 5 hPa, the thick line is the 990 hPa contour), equivalent potential temperature (filled contours, values shown by the colour bar at the bottom), and winds (maximum wind vector is 17 m/s) simulated at 25 m above the underlying surface in the terrain-following coordinates of RAMS. This figure is derived from the RAMS run at 10 km horizontal-resolution and shows the domain covered by this run. The bottom and left axes show the grid point number, while the top and right axes show the geographical coordinates.








**a)**

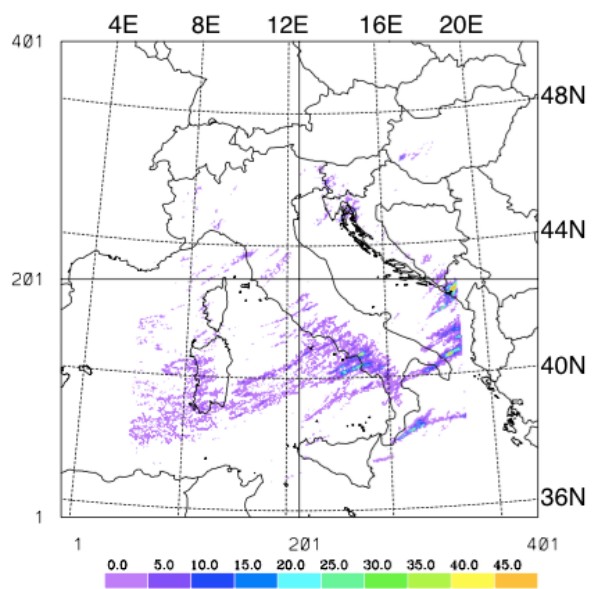



**b)**

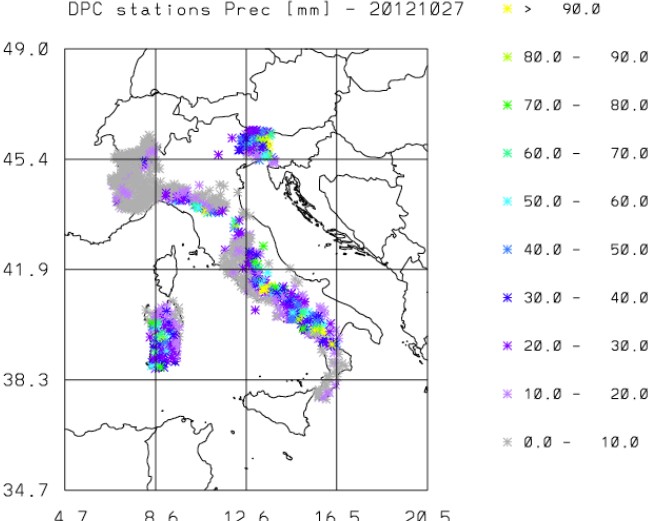


Figure 3: a) Lightning density on 27 October 2012 [number of flashes/16 km$^2$]. The lightning
number is obtained by remapping the lightning observed by LINET onto the RAMS grid at 4 km
horizontal resolution. Note that the lightning are cut on all sides (this is especially evident on the
Eastern bound) because of the data availability. The figure shows the RAMS domain for R4. The
bottom and left axes show the grid point number, while the top and right axes show the
geographical coordinates; b) daily precipitation [mm] recorded by available raingauges on 27
October 2012.




**a)**

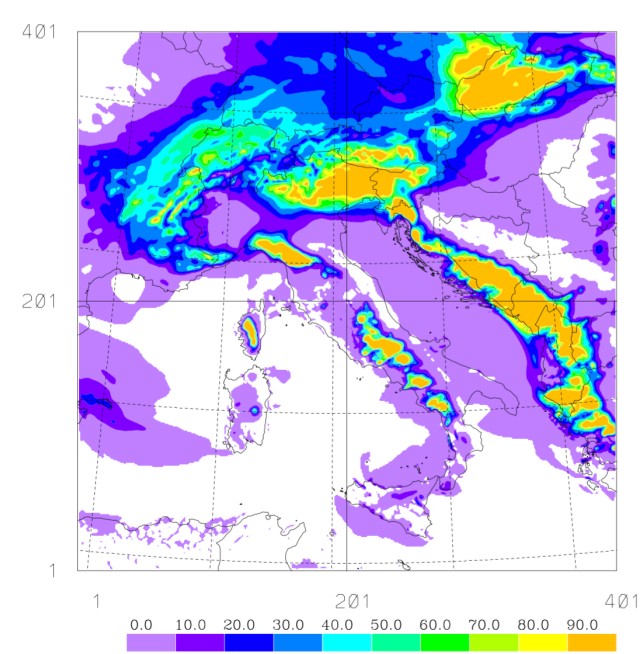



**b)**

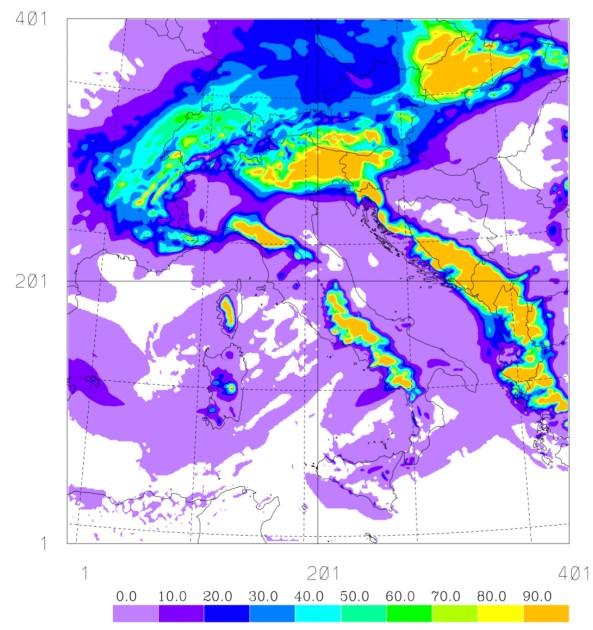





**c)**

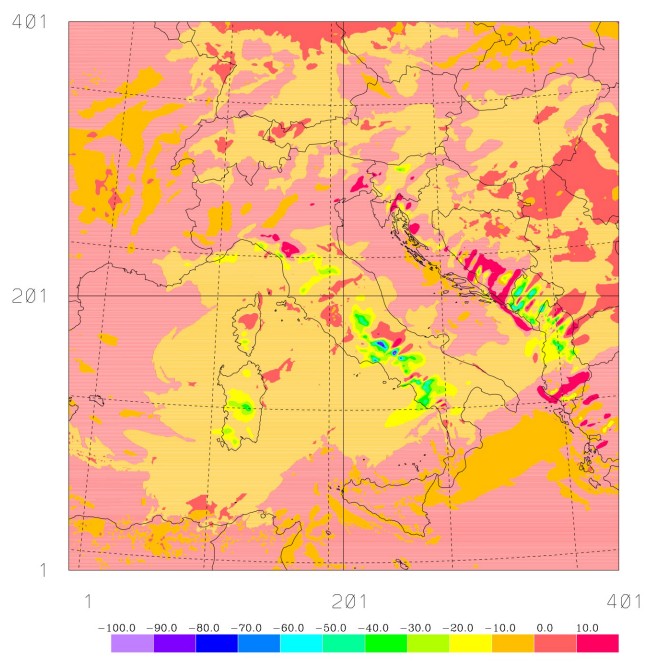


Figure 4 a) daily precipitation [mm] forecast of CNTRL; b) daily precipitation [mm] forecast
obtained by summing the eight 3h forecasts of F3HA6; c) difference of daily precipitation [mm]
between CNTRL and F3HA6.














**a)**

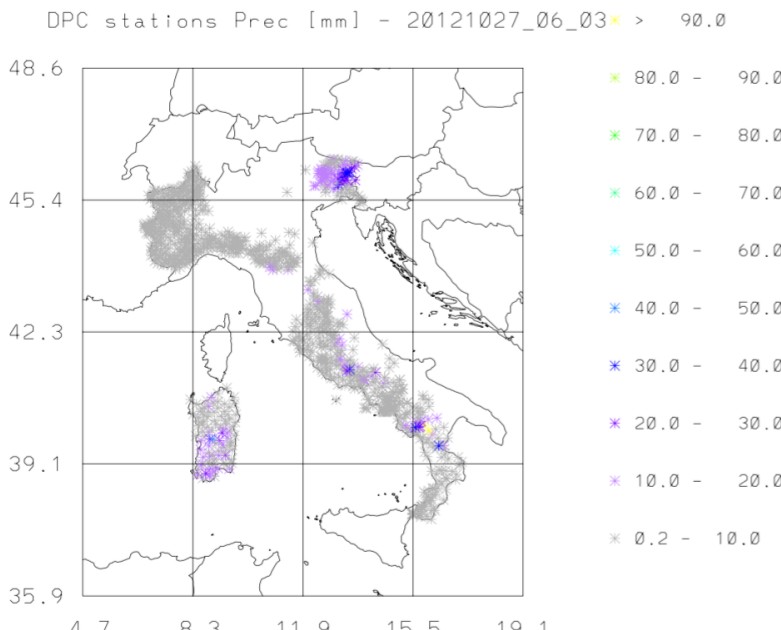



**b)**

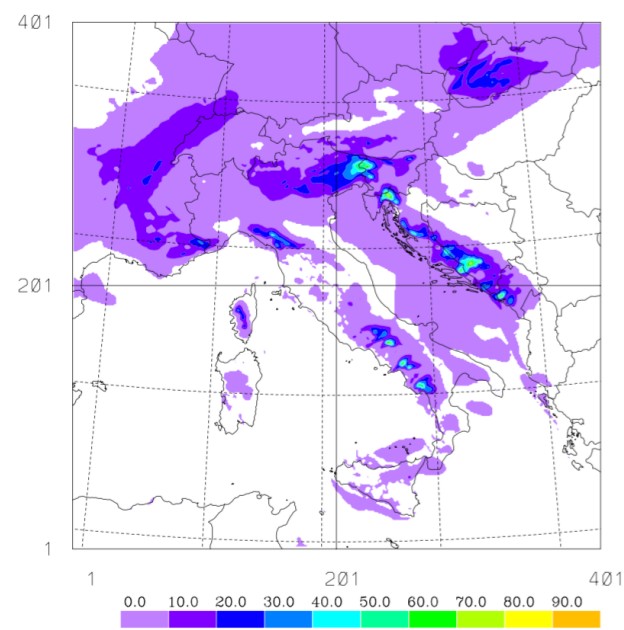




**c)**

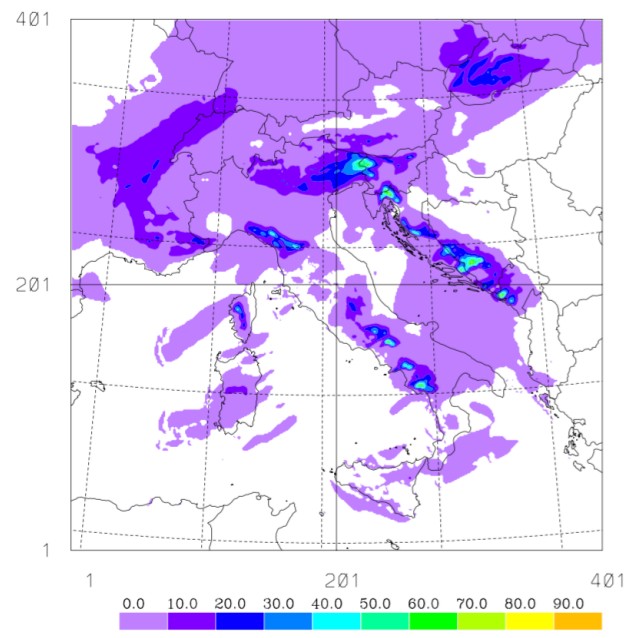



Figure 5: a) Precipitation [mm] recorded by raingauges between 06 and 09 UTC; b) As in a) for the
CNTRL forecast; c) As in a) for the F3HA6 forecast.


















**a)**

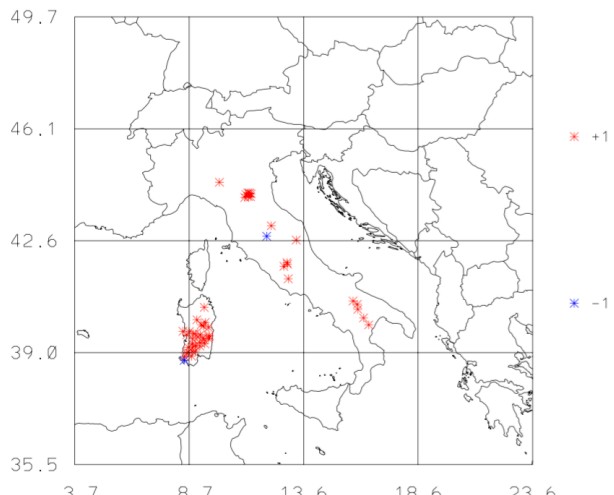



**b)**

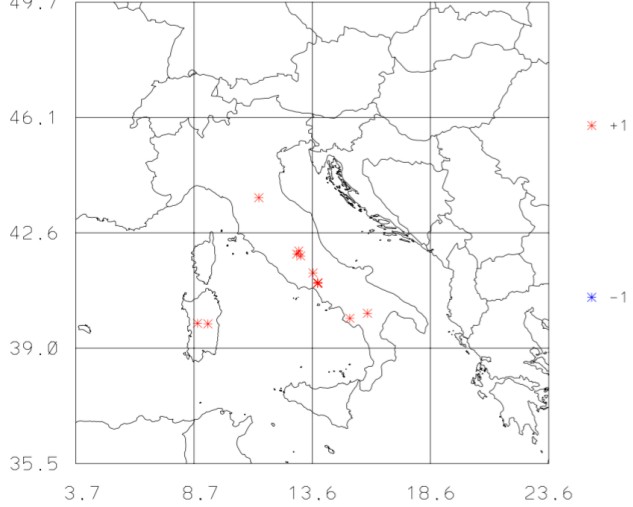


Figure 6: a) Difference between the hits of the contingency tables of F3HA6 and CNTRL for the 1
mm/3h (8 mm/day) forecast; b) As in a) for the 10 mm/3h (80 mm/day) threshold.















Figure 7: Scores for the daily precipitation computed by summing the contingency tables of all
twenty case studies; a) Bias (the line of the perfect score 1.0 is shown in black); b) Equitable Threat
Score; c) Probability of Detection; d) False Alarm Ratio. F3HA6 is in green, ASSIM is in red and
CNTRL in blue. The asterisks above the x-axis show the results of the hypothesis testing (95%
blue, 90% red) of the difference between F3HA6 and CNTRL scores.














Figure 8: Scores for the 3h precipitation computed by summing the 160 contingency tables of the
twenty case studies; a) Bias (the line of the perfect score 1.0 is shown in black); b) Equitable Threat
Score; c) Probability of Detection; d) False Alarm Ratio. F3HA6 is in green, ASSIM is in red and
864       CNTRL in blue. The asterisks above the x-axis show the results of the hypothesis testing (95%
865              blue, 90% red) of the difference between F3HA6 and CNTRL scores.
