# Peer review of "Improvement of RAMS precipitation forecast at the short range through lightning data assimilation"

_Natural Hazards and Earth System Sciences, 2016_

## Referee Comment (RC1) · Anonymous Referee #1 · 1 Oct 2016

The paper describes the application of a methodology for the assimilation of lightning data into RAMS in 20 case studies characterized by widespread convection and lightning activity. First, the analysis focuses on a case study of intense convection during the HyMeX SOP1 campaign, then statistical indices are derived for all the cases analyzed. Results show a clear improvement due to use of assimilation technique compared to the control run (without assimilation). The paper is well written and appropriate for NHESS, thus I recommend publication after minor revisions.

MINOR POINTS: Line 120: why did you choose 4 km as inner grid spacing? This corresponds to the grey area for convection and it is slightly below actual standards (2-3 km). For future studies, I suggest to test your assimilation technique at higher

resolution; Line 181: I understand you increased the water content only in the charged zone (0°C - -25°C): is there a relaxation region above and below this area, or did you just change the values only in that zone? In the latter case, did you notice whether the discontinuity in water vapor generated a perturbation affecting the lower and upper regions? Line 213: please write explicitly that the "previous R4 forecast" belongs to the F3HA6 set of simulations; Lines 216-217: please change into "Please note the switch of the initial conditions ...”; Lines 266-281: I suggest to remove this part from here and put in a specific Appendix, possibly explaining the resampling technique more in detail; Line 306: please change into "From Fig. 3a, convection is apparent over the Tyrrhenian Sea and is enhanced over land because of ...”; Lines 319: "for the largest threshold": do you mean "above 90 mm/day"? Line 355: delete "a" or change "spells" in singular; Line 385: in how many stations was the precipitation "subtracted where it did not occur"? Line 399: "... increases with the threshold from ...”; Figure 7: since the lower threshold you consider is 1 mm/day, I believe showing also 0 mm in the x-axis is not proper; Lines 436-441: the assimilation increases the rainfall amount, thus the hit rate and POD are better, but there is a general overestimation (thus, the bias is higher and there is an increase of false alarms). Anyway, I agree with you that, even with these limitations, the result is overall helpful for operational purposes. I suggest you should speculate more on this point; Lines 442-462: the description of Fig. 8 is too long: you can reduce this part referring to the similarities with Fig. 7; Line 475 and elsewhere: convection without "the"; Lines 474-479: are the results for the other cases similar to those for October 27? Line 511: "... improvement in some statistical scores, ...”; Line 519: please rephrase into "... the performance of the precipitation forecast is clearly dependent on the type of event ...”; Figure 3: apparently, the maximum threshold of 90 mm is too small, thus the peak in simulated rainfall cannot be clearly estimated; please, could you add the information about the maximum precipitation simulated by the model at least in the text?

[Figure]

2016.

---

## Referee Comment (RC2) · Anonymous Referee #2 · 9 Oct 2016

Overall evaluation The paper presents the evaluation of a lightning data assimilation, implemented in the RAMS model. Overall, the manuscript is well written and the methodology and results are well discussed relative to the available international literature. The subject is of high interest. I suggest acceptance of the manuscript, subject to some minor comments and technical corrections, which are summarized in the following.

Minor comments 1. L64-76: I believe that the three paragraphs could be merged in one, as they all present briefly examples of lightning data assimilation studies. 2. I suggest that Table 1 is removed from the manuscript. Instead of presenting this detailed information on the domain configuration of RAMS, it could be of more usefulness for

the reader to add a simple plot showing the domains or stick to the two figures that are already referenced for giving an overview on domains. The respective description of the domain configuration in L120-126 can remain as is. 3. L123: It would be also proper to include model top in hPa. 4. L151-152: I believe that simply referring to cloud-to-ground (CG) and intra-cloud (IC) lightning is enough, instead of giving the information currently shown in the parentheses. 5. L223-227: This particular part of the manuscript presents a result of the study. Hence, it can be removed from Sec. 2, that is devoted to methodology. It can be moved to the Results section, at the appropriate place. 6. L259-L263: I think that this paragraph could be removed as it reports information that is most probably well known to the interested readers. I leave it up to the authors to decide whether it should be removed or stay. 7. L290-293: Please rewrite this part of the manuscript in a more clear way.

Technical corrections 1. L1: "short-range" 2. L18: "set up" 3. L21, L78, L95, L190: "that occurred" 4. L21: "which were..." 5. L23, L244, L313 and throughout the entire manuscript: "rain gauges" 6. L23, L95: "target region" or "target area" 7. L73: "presented" or "introduced" could be a better choice for this sentence. 8. L75: "performed" instead of "made". 9. L131: The correct terminology is "WRF single-moment six-class microphysics scheme" 10. L174, L322: "setup". 11. L174: "trial and error analysis..." 12. L244: The abbreviation QPF (Quantitative Precipitation Forecast) has not been previously defined, should I have not missed it while reading the manuscript. 13. L249: "with" instead of "being". 14. L267, L484: "competing" instead of "competitor". 15. L298-290: Correct the term to "troughs" (it is now written as "through"). 16. L290: "cut-off low". 17. L495: "WRF-LTNGDA"

---

## Author Comment (AC1) · 21 Nov 2016

Overall evaluation: The paper presents the evaluation of a lightning data assimilation, implemented in the RAMS model. Overall, the manuscript is well written and the methodology and results are well discussed relative to the available international lit- erature. The subject is of high interest. I suggest acceptance of the manuscript, sub- ject to some minor comments and technical corrections, which are summarized in the following.

Minor comments

1. L64-76: I believe that the three paragraphs could be merged in one, as they all present briefly examples of lightning data assimilation studies.

   - The three paragraphs will be merged in the revised version of the paper.

2. I suggest that Table 1 is removed from the manuscript. Instead of presenting this detailed information on the domain configuration of RAMS, it could be of more usefulness for the reader to add a simple plot showing the domains or stick to the two figures that are already referenced for giving an overview on domains. The respective description of the domain configuration in L120-126 can remain as is.

-Thank you for this comment, we will follow this suggestion including a new Figure 1, showing the domains. Table 1 will be removed and the Figure 1 caption will include some details on the domains.

Here is the Figure 1:

[Figure]

Figure 1: The two domains (D1, D2). D1 has 301 grid points in both the WE and SE directions; D2 has 401 grid points in both WE and SE directions.

3. L123: It would be also proper to include model top in hPa.

- In the RAMS model, the model top is fixed in z, the height above the sea level. Pressure varies on the model top surface, however an estimation of the model top in hPa (40 hPa), from the model output will be given.

4. L151-152: I believe that simply referring to cloud-to- ground (CG) and intra-cloud (IC) lightning is enough, instead of giving the information currently shown in the parentheses.

-We will change the paper according to the comment.

5. L223-227: This particular part of the manuscript presents a result of the study. Hence, it can be removed from Sec. 2, that is devoted to methodology. It can be moved to the Results section, at the appropriate place.

- This part will be moved in the Results section.

6. L259-L263: I think that this paragraph could be removed as it reports information

that is most probably well known to the interested readers. I leave it up to the authors to decide whether it should be removed or stay.

-We agree with reviewer that this part of the paper could be removed because it presents basic definition well known to the large part of readers. Nevertheless, in other papers, we found that reviewers asked for this explanation about the scores and, in the doubt, we will maintain the paragraph.

7. L290-293: Please rewrite this part of the manuscript in a more clear way.

Thank you for noting this point. We will clarify the sentence: "During SOP1, several upper level troughs extended from the Northern and Central Europe toward the Mediterranean Basin or entered in the Basin as deep trough. Few of them developed a cut-off low at 500 hPa; the interaction between the upper level troughs and the orography of the Alps generated a low pressure pattern at the surface in Northern Italy, and usually the whole system moved along the Italian peninsula. The 27 October 2012 case study, also referred as IOP16a, belongs to this class of events, but it eventually evolved in a cut-off at 500 hPa on 28-29 October (IOP16c)."

**Technical corrections:**

All the technical points below will be corrected according to the comments.

1. L1: "short-range"

2. L18: "set up"

3. L21, L78, L95, L190: "that occurred"

4. L21: "which were..."

5. L23, L244, L313 and throughout the entire manuscript: "rain gauges"

6. L23, L95: "target region" or "target area"

7. L73: "presented" or "introduced" could be a better choice for this sentence.

8. L75: "performed" instead of "made".

9. L131: The correct terminology is "WRF single-moment six-class microphysics scheme"

10. L174, L322: "setup".

11. L174: "trial and error analysis..."

12. L244: The abbreviation QPF (Quantitative Precipitation Forecast) has not been previously defined, should I have not missed it while reading the manuscript.

13. L249: "with" instead of "being".

14. L267, L484: "competing" instead of "competitor".

15. L298-290: Correct the term to "troughs" (it is now written as "through").

16. L290: "cut-off low".

17. L495: "WRF-LTNGDA"

---

## Author Comment (AC2) · 2 Dec 2016

Reviewer #1

The paper describes the application of a methodology for the assimilation of lightning data into RAMS in 20 case studies characterized by widespread convection and lightning activity. First, the analysis focuses on a case study of intense convection during the HyMeX SOP1 campaign, then statistical indices are derived for all the cases an- alyzed. Results show a clear improvement due to use of assimilation technique com- pared to the control run (without assimilation). The paper is well written and appropriate for NHESS, thus I recommend publication after minor revisions.

Line 120: why did you choose 4 km as inner grid spacing? This corresponds to the grey area for convection and it is slightly below actual standards (2-3 km). For future studies, I suggest to test your assimilation technique at higher resolution;

-This point is of great interest because of the important role that the horizontal resolution plays in mesoscale models, including the impact that the horizontal resolution has on the resolved vs not resolved, i.e. convective, precipitation. The reason for choosing 4 km horizontal resolution is motivated by operational reasons. The methodology of this paper is implemented in a real-time weather forecasting system at ISAC-CNR and we study the performance of this specific system. A finer horizontal resolution cannot be implemented operationally with the current computing power.

Nevertheless, the impact of the horizontal resolution is notable. To better quantify this point we increased the horizontal resolution from 4 km (the resolution of the paper) to 2.5 km for the 15 October 2012 and 27 October case studies.

Figures 1 and 2 show the precipitation of the simulations F3HA6 (the assimilation scheme was tuned for the new resolution as stated in the paper) for the 27 October case study. The impact of the resolution is notable because the precipitation patterns, especially at high thresholds (>50 mm/day), are less spread in the 2.5 km horizontal resolution experiment. This behavior is apparent all along the Apennines, but it is especially important in NE Italy, where the precipitation area for thresholds larger than 90 mm is reduced in the 2.5 km horizontal resolution forecast compared to 4 km. The impact could be beneficial for the scores of the F3HA6 because it has the tendency to overestimate the precipitation, especially at high thresholds. A similar behavior was found for the CNTRL forecast of the 27 October (not shown).

[Figure]

Figure 1: Precipitation [mm] accumulated for the 27 October 2015 for the simulation F3HA6 at 2.5 km horizontal resolution.

[Figure]

Figure 2: Precipitation [mm] accumulated for the 27 October 2015 for the simulation F3HA6 at 4 km horizontal resolution (as in the paper, Figure 5b).

For the 15 October case, the Figures 3 and 4 show the results of the CNTRL forecast, i.e. without lightning assimilation.

[Figure]

Figure 3: Precipitation [mm] accumulated for the 27 October 2015 for the simulation CNTRL at 2.5 km horizontal resolution.

[Figure]

Figure 4: Precipitation [mm] accumulated for the 15 October 2015 for the simulation CNTRL at 4km horizontal resolution.

Also for this case study, it is apparent the less spread of the precipitation for the larger threshold (>50 mm/day) both over NE of Italy and over the Apennines for the 2.5 km horizontal resolution, caused by the more realistic representation of the interaction between air masses and local orography. A similar behavior was found comparing F3HA6 forecast at 4 km and 2.5 km horizontal resolutions for the 15 October case study (not shown).

Considering the number of stations where the forecast is improved by the lightning data assimilation we noted again an important improvement at 2.5 km horizontal resolution. Nevertheless, the number of stations where the forecast is improved by the lightning data assimilation decreases in the 2.5 km horizontal resolution compared to 4 km horizontal resolution. This is mainly caused by an improvement of the CNTRL forecast at 2.5 km compared to that at 4 km. Also, the number of false alarms are reduced at finer horizontal resolution.

It is finally noted that the above results, while interesting, are preliminary and will be further investigated in future studies.

A discussion about this point will be included at the end of section "4. Discussion and conclusions", summarizing the above considerations.

We will write: "Finally, horizontal resolutions higher than that of this paper are needed to better resolve the orography and its interaction with air masses. To quantify this point preliminary, we increased the horizontal resolution of the second domain from 4 km to 2.5 km for the 15 October and 27 October case studies. Results for the two cases show that the impact of the horizontal resolution is notable because the precipitation patterns, especially for larger thresholds (>50 mm/day), are less spread at 2.5 km horizontal resolution compared to 4 km forecast (see the discussion of this paper for the daily precipitation maps for the two cases, Federico et al., 2016). This impact could be beneficial for the scores of the F3HA6 forecast because it has the tendency to overestimate the precipitation area at high thresholds, as shown in this paper. However, these results are preliminary, and future studies are needed to quantify the important impact of the horizontal resolution on the lightning data assimilation forecast."

Also, at the end of section 2.1 we will write: "Before concluding this section it is important to note that 4 km horizontal resolution of the finer grid corresponds to the grey area for convection and it is slightly below actual standards (2-3 km). This resolution was motivated by operational purposes: the methodology of this

paper is implemented in a real-time weather forecasting system at ISAC-CNR and we study the performance of this specific system. Preliminary results of the impact of the horizontal resolution on the lightning assimilation are discussed in Section 4."

Line 181: I understand you increased the water content only in the charged zone (0°C - -25°C): is there a relaxation region above and below this area, or did you just change the values only in that zone? In the latter case, did you notice whether the discontinuity in water vapour generated a perturbation affecting the lower and upper regions?

-We change the water vapour in the charging zone between 0°C and -25°C, without relaxing zone. The water vapour, however, is redistributed by the model advection/diffusion and is changed also outside the charging zone.

Figure 5 shows the difference between the water vapour mixing ratio at 760 m above the ground level in the terrain following coordinate system of RAMS for the 22 UTC of 15 October 2012.

[Figure]

Figure 5: Difference [g/kg] of the water vapor at 760 m in the terrain-following coordinate system of RAMS of the F3HA6 and CNTRL simulations at 22 UTC on 15 October 2012.

Differences are well evident over the Tyrrhenian Sea. Over the sea, the 760 m level is well below the charging zone (0; -25°C; roughly between 3000 and 6500 m a.s.l. for the time shown in Figure 5), showing the impact of the lightning data assimilation on the water vapour distribution outside the charging zone.

During the hour preceding the time of Figure 5, 3315 were observed by LINET (Figure 6). Most of them are over Sicily showing the direct effect of lightning in the redistribution of the water vapour. Also, the differences over the Tyrrhenian Sea North of 40°N are mainly caused by the differences in the storm evolutions of the two simulations.

[Figure]

Figure 6: Flashes observed between 21 and 22 UTC on 15 October 2012.

We will add a sentence about this point in the Section "*2.2 Lightning data and assimilation procedure*".

We will write: "It is noted that we change the water vapour in the charging zone between 0°C and -25°C, without a relaxing zone. The water vapour, however, is redistributed by the model advection, diffusion and diabatic processes, and it is changed also outside the charging zone (see the discussion of this paper; Federico et al. 2016)."

Line 213: please write explicitly that the "previous R4 forecast" belongs to the F3HA6 set of simulations;

-Done. We will write: "The second F3HA6 simulation starts at 21 UTC of the day before the actual day using as initial conditions the previous R4 forecast, belonging to F3HA6 set of simulations, and as BC the R10 forecast."

Lines 216-217: please change into "Please note the switch of the initial conditions ...";

-This sentence will be changed.

Lines 266-281: I suggest to remove this part from here and put in a specific

Appendix, possibly explaining the resampling technique more in detail;

- We will move this specific part of the paper to the Appendix A and we will extend the discussion on the statistical test in this Appendix. Here is the new Appendix A:

"Appendix A

We use the resampling method introduced by Hamill (1999) for the comparison of the scores of CNTRL and F3HA6 forecasts (see also Accadia et al. (2003) and Federico et al. (2003)).

The null hypothesis is that the difference of the scores of the two competing models, CNTRL and F3HA6, is zero:

$$H_0: \ S_1\text{-}S_2=0 \tag{A1}$$

Where $S$ is the generic score (Bias, ETS, POD and FAR), 1 is the CNTRL forecast and 2 is the F3HA6 forecast. The scores are computed from the sum of the contingency tables of the CNTRL and F3HA6 forecasts to minimize the sensitivity of the test to small changes of the contingency table elements.

In this paper the number of contingency tables available is 8 multiplied the number of days, i.e. $n=20*8=160$ for the 3h precipitation forecast, and $n=20$ for the daily precipitation forecast. Indicating the contingency tables by the vector $\boldsymbol{x}$:

$$\boldsymbol{x}_{i,j}=(a,b,c,d)_{i,j} \tag{A2}$$

where $i$ is the competing model ($i=1$ for CNTRL, $i=2$ for F3HA6) and $j$ is the contingency table ($j=$ 1,…,180 for 3h forecast, and $j=1,…,20$ for daily precipitation), the scores are computed from the sum of the contingency tables:

$$S_i = f\left(\sum_{j=1}^{n} x_{i,j}\right) \tag{A3}$$

and the test statistic is given by the difference between $S_1$ and $S_2$.

The bootstrap method is applied by resampling the contingency tables in a consistent way. For this purpose, a random number $I_j$ is generated, which can assume the values 1 or 2. If $I_j$ is 1 the contingency table of CNTRL is selected, if $I_j = 2$ the F3HA6 table is selected. The process is repeated for each contingency table ($j=1,…,180$ for 3h forecast, and $j=1,…,20$ for daily precipitation) and the scores $S_1^*$ and $S_2^*$ are computed:

$$S_1^* = f\left(\sum_{j=1}^{n} x_{I_j,j}\right); \ \ S_2^* = f\left(\sum_{j=1}^{n} x_{3-I_j,j}\right) \tag{A4}$$

So, the two $j$-th contingency tables are swapped if $I_j=2$, while the swapping is not

performed for $I_j=1$.

This random sampling is performed a large number of times (10.000 in this paper). Each time the scores are computed from the sum of the elements of the resampled contingency tables, Eqn. (A4), to make the null distribution $(S_1^*-S_2^*)$ of the difference between the scores of the competing forecasts.

Then we compute the $t_L$ and $t_U$ that represent the $\alpha/2$ and $(1-\alpha)/2$ percentile of the null distribution $(S_1^*-S_2^*)$. The null hypothesis that the score difference between the two competing forecasts is zero is rejected at the level 90 % ($\alpha =0.1$) or 95% ($\alpha =0.05$) if:

$$(S_1- S_2) < t_L \quad \text{or} \quad (S_1- S_2) > t_U \tag{A5}$$

where $S_1$ and $S_2$ are the generic scores of the actual distributions (not resampled)."

Line 306: please change into "From Fig. 3a, convection is apparent over the Tyrrhenian Sea and is enhanced over land because of . . .";

-Ok.

Lines 319: "for the largest threshold": do you mean "above 90 mm/day"?

-We will change the sentence to be more clear: "However, the precipitation is overestimated by both CNTRL and F3HA6, especially above 30 mm/day."

Line 355: delete "a" or change "spells" in singular;

-ok.

Line 385: in how many stations was the precipitation "subtracted where it did not occur"?

- In the revised version of the paper, this will be quantified by counting the number of stations where the precipitation is lowered by at least 1 mm/3h (110 stations), 5 mm/3h (20 stations), and 10 mm/3h (7 stations) between the 03 and 06 UTC of 27 October, when the lightning data assimilation is used. We will write: " For example, between 03 and 06 UTC there are 110 stations where the precipitation is reduced by more than 1 mm/3h, 20 stations where it is reduced by more than 5 mm/3h and 7 stations for which the precipitation is reduced by more than 10 mm/3h."

Line 399: ". . . increases with the threshold from . . ."; Figure 7: since the lower

threshold you consider is 1 mm/day, I believe showing also 0 mm in the x-axis is not proper;

- The reviewer is referring to Figures 8 and 9 of the revised version of the paper. We will redraw these figures (8 and 9) according to this comment.

Lines 436-441: the assimilation increases the rainfall amount, thus the hit rate and POD are better, but there is a general overestimation (thus, the bias is higher and there is an increase of false alarms). Anyway, I agree with you that, even with these limitations, the result is overall helpful for operational purposes. I suggest you should speculate more on this point;

-Thank you for suggesting this point. We will write: "The inspection of the contingency tables shows that the improvement of the FAR for those thresholds is attained by a larger number of hits but there is also an increase of the false alarms. In general, the lighting assimilation increases the precipitation, which is already overestimated for the larger thresholds by CNTRL. So, the POD and the hit rate are increased by lightning data assimilation, but also the false alarms, which were already reported in CNTRL, especially for larger thresholds. Anyway, we believe that the result is overall helpful for operational purposes."

Lines 442-462: the description of Fig. 8 is too long: you can reduce this part referring to the similarities with Fig. 7;

-The discussion of Figure 8 (Figure 9 in the revised paper) will be shortened.

Line 475 and elsewhere: convection without "the";

-This will be corrected.

Lines 474-479: are the results for the other cases similar to those for October 27?

-The impact of the lightning data assimilation on convection over the sea is significant and has an important role in most cases. For example, a similar behaviour to the 27 October was found for the 15 October and 12 October case studies with impacts on the Tuscany and Lazio regions, i.e. the central Western coast of the Italian peninsula. Other cases are evident in the Western coast of Southern Italy (for example the 31 October 2012 but also others). There are occasions, however, where convection over the Sea is less important. For example, the 12 September was characterized by a severe storm over Friuli Venezia Giulia (Manzato et al., 2014). For this case study, the difference between the precipitation of the CNTRL forecast and that of F3HA6 (i.e. the lightning assimilation forecast), in this order, is shown below:

[Figure]

Figure 7: Difference of daily precipitation [mm] between CNTRL and F3HA6 for the 12 September case study.

In this case, the difference is confined over the land (NE of Italy), and the role of convection over the sea is less important, at least as the initiation mechanism for convection over land. However, air masses advected from the Adriatic Sea toward the storm centre play an important role in feeding the storm with latent heat. We will add a comment about this point in Section "4 Discussion and conclusions".

We will write: "The advection of convection from the sea to the land was important in most case studies considered in this paper, and we can conclude that it plays a fundamental role. There are cases, however, when it is less important, as for the severe and localized storm that occurred in NE Italy on 12 September 2012 (Manzato et al., 2014). For this case, the storm developed and evolved over land, and the difference between the precipitation fields of the CNTRL and F3HA6 is confined inland, over NE Italy, and it is larger than 40 mm (see the discussion of this paper for the map of the precipitation difference between CNTL and F3HA6; Federico et al., 2016)."

Line 511: ". . . improvement in some statistical scores, . . .";

-Ok.

Line 519: please rephrase into ". . . the performance of the precipitation forecast is clearly dependent on the type of event . . .";

-This sentence will be rephrased.

Figure 3: apparently, the maximum threshold of 90 mm is too small, thus the peak in simulated rainfall cannot be clearly estimated; please, could you add the information about the maximum precipitation simulated by the model at least in the text?

-The Figure 3 is Figure 4 in the revised paper. We will add this information in the Figure 4 caption (the maximum value is 320 mm in Southern Italy; over NE Italy the maximum simulated value is 132 mm). Also, we will add the largest value observed in the text, when commenting Figure 4b. We will write: "The largest precipitation recorded in NE Italy is 141 mm (13.54E, 45.85N), while more than 200 mm are reported in two stations in Southern Italy (15.84E, 40.31N; 207 mm) and (15.98E, 40.16N; 220 mm).".

References (added to the paper):

Manzato, A., S. Davolio, M. M. Miglietta, A. Pucillo, and M. Setvák, 2014: 12 September 2012: A supercell outbreak in NE Italy?. Atmos. Res., 153, 98-118.

Federico, S., Petracca, M., Panegrossi, G., and Dietrich, 2016: Improvement of RAMS precipitation forecast at the short-range through lightning data assimilation. Nat. Hazards Earth Syst. Sci. Discuss., doi:10.5194/nhess-2016-291.

---

## Author Response (AR1)

Answer to the reviewers' comments

Dear Editor,

We thank both reviewers for the instructive comments, that improved the quality
of the paper. We also thank the Editor for the support in managing the discussion
of this paper. In the following, there are our detailed answers to the reviewers'
comments. Also, we added the following two references.

Manzato, A., S. Davolio, M. M. Miglietta, A. Pucillo, and M. Setvák, 2014: 12
September 2012: A supercell outbreak in NE Italy?. Atmos. Res., 153, 98-118.

Federico, S., Petracca, M., Panegrossi, G., and Dietrich, 2016: Improvement of
RAMS precipitation forecast at the short-range through lightning data
assimilation. Nat. Hazards Earth Syst. Sci. Discuss., doi:10.5194/nhess-2016-291.

The marked-up manuscript version of the paper, containing all the changes we did,
follows the answer to the reviewer comments. The Italian word "Eliminato" means
"Deleted".

Reviewer #1

The paper describes the application of a methodology for the assimilation of
lightning data into RAMS in 20 case studies characterized by widespread
convection and light- ning activity. First, the analysis focuses on a case study of
intense convection during the HyMeX SOP1 campaign, then statistical indices are
derived for all the cases an- alyzed. Results show a clear improvement due to use
of assimilation technique com- pared to the control run (without assimilation). The
paper is well written and appropriate for NHESS, thus I recommend publication
after minor revisions.

Line 120: why did you choose 4 km as inner grid spacing? This corresponds to the
grey area for convection and it is slightly below actual standards (2-3 km). For
future studies, I suggest to test your assimilation technique at higher  resolution;

-This point is of great interest because of the important role that the horizontal
resolution plays in mesoscale models, including the impact that the horizontal
resolution has on the resolved vs not resolved, i.e. convective, precipitation. The
reason for choosing 4 km horizontal resolution is motivated by operational
reasons. The methodology of this paper is implemented in a real-time weather forecasting system at ISAC-CNR and we study the performance of this specific system. A finer horizontal resolution cannot be implemented operationally with the current computing power.

Nevertheless, the impact of the horizontal resolution is notable. To better quantify this point we increased the horizontal resolution from 4 km (the resolution of the paper) to 2.5 km for the 15 October 2012 and 27 October case studies. The figures of the precipitation fields with or without lightning data assimilation at 4 and 2.5 km horizontal resolution have been shown in the discussion on this paper with Reviewer #1. These preliminary results show that the impact of the horizontal resolution is notable because the precipitation patterns, especially for larger thresholds (>50 mm/day), are less spread at 2.5 km horizontal resolution compared to 4 km forecast.

We wrote: "Finally, horizontal resolutions higher than that of this paper are needed to better resolve the orography and its interaction with air masses. To quantify this point preliminary, we increased the horizontal resolution of the second domain from 4 km to 2.5 km for the 15 October and 27 October case studies. Results for the two cases show that the impact of the horizontal resolution is notable because the precipitation patterns, especially for larger thresholds (>50 mm/day), are less spread at 2.5 km horizontal resolution compared to 4 km forecast (see the discussion of this paper for the daily precipitation maps for the two cases, Federico et al., 2016). This impact could be beneficial for the scores of the F3HA6 forecast because it has the tendency to overestimate the precipitation area at high thresholds, as shown in this paper. However, these results are preliminary, and future studies are needed to quantify the important impact of the horizontal resolution on the lightning data assimilation forecast."

Also, at the end of section 2.1 we wrote: "Before concluding this section it is important to note that 4 km horizontal resolution of the finer grid corresponds to the grey area for convection and it is slightly below actual standards (2-3 km). This resolution was motivated by operational purposes: the methodology of this paper is implemented in a real-time weather forecasting system at ISAC-CNR and we study the performance of this specific system. Preliminary results of the impact of the horizontal resolution on the lightning assimilation are discussed in Section 4."

Line 181: I understand you increased the water content only in the charged zone (0°C - -25°C): is there a relaxation region above and below this area, or did you just change the values only in that zone? In the latter case, did you notice whether the discontinuity in water vapor generated a perturbation affecting the lower and upper regions?

-We change the water vapour in the charging zone between 0°C and -25°C, without relaxing zone. The water vapour, however, is redistributed by the model advection/diffusion and it is changed also outside the charging zone.

An example of this behaviour has been shown in the discussion of this paper with Reviewer #1.

We added a sentence about this point in the Section "*2.2 Lightning data and assimilation procedure*".

We wrote: "It is noted that we change the water vapour in the charging zone between 0°C and -25°C, without a relaxing zone. The water vapour, however, is redistributed by the model advection, diffusion and diabatic processes, and it is changed also outside the charging zone (see the discussion of this paper; Federico et al. 2016)."

Line 213: please write explicitly that the "previous R4 forecast" belongs to the F3HA6 set of simulations;

- We wrote: "The second F3HA6 simulation starts at 21 UTC of the day before the actual day using as initial conditions the previous R4 forecast, belonging to F3HA6 set of simulations, and as BC the R10 forecast."

Lines 216-217: please change into "Please note the switch of the initial conditions ...";

-Done

Lines 266-281: I suggest to remove this part from here and put in a specific Appendix, possibly explaining the resampling technique more in detail;

-Done. We moved this specific part to the Appendix A and we extended the discussion.

Line 306: please change into "From Fig. 3a, convection is apparent over the Tyrrhenian Sea and is enhanced over land because of . . .";

-Done.

Lines 319: "for the largest threshold": do you mean "above 90 mm/day"?

-We changed the sentence to be clearer: "However, the precipitation is
overestimated by both CNTRL and F3HA6, especially above 30 mm/day."

Line 355: delete "a" or change "spells" in singular;

-We deleted "a"

Line 385: in how many stations was the precipitation "subtracted where it did not
occur"?

- In the revised version of the paper, this has been quantified by counting the
number of stations where the precipitation is lowered by at least 1 mm/3h (110
stations), 5 mm/3h (20 stations), and 10 mm/3h (7 stations) between the 03 and 06
UTC of 27 October, when the lightning data assimilation is used. We wrote: " For
example, between 03 and 06 UTC there are 110 stations where the precipitation is
reduced by more than 1 mm/3h, 20 stations where it is reduced by more than 5
mm/3h and 7 stations for which the precipitation is reduced by more than 10
mm/3h."

Line 399: ". . . increases with the threshold from . . ."; Figure 7: since the lower
threshold you consider is 1 mm/day, I believe showing also 0 mm in the x-axis is
not proper;

The reviewer is referring to Figures 8 and 9 of the revised version of the paper.
We changed these figures (8 and 9) according to this comment.

Lines 436-441: the assimilation increases the rainfall amount, thus the hit rate and
POD are better, but there is a general overestimation (thus, the bias is higher and
there is an increase of false alarms). Anyway, I agree with you that, even with
these limitations, the result is overall helpful for operational purposes. I suggest
you should speculate more on this point;

-Thank you for suggesting this point. We wrote: "The inspection of the
contingency tables shows that the improvement of the FAR for those thresholds is
attained by a larger number of hits but there is also an increase of the false alarms.
In general, the lighting assimilation increases the precipitation, which is already
overestimated for the larger thresholds by CNTRL. So, the POD and the hit rate are increased by lightning data assimilation, but also the false alarms, which were already reported in CNTRL because of the general overestimation of the rainfall. Anyway, we believe that the result is overall helpful for operational purposes."

Lines 442-462: the description of Fig. 8 is too long: you can reduce this part referring to the similarities with Fig. 7;

-The discussion was shortened.

Line 475 and elsewhere: convection without "the";

-Corrected

Lines 474-479: are the results for the other cases similar to those for October 27?

-The impact of the lightning data assimilation on convection over the sea is significant and has an important role in most cases. For example, a similar behaviour to the 27 October was found for the 15 October and 12 October case studies with impacts on the Tuscany and Lazio regions, i.e. the central Western coast of the Italian peninsula. Other cases are evident in the Western coast of Southern Italy (for example the 31 October 2012 but also others). There are occasions, however, where convection over the Sea is less important. For example, the 12 September was characterized by a severe storm over Friuli Venezia Giulia (Manzato et al., 2014). In this case, the difference is confined over the land (NE of Italy), and the role of convection over the sea is less important, at least as the initiation mechanism for convection over land. However, air masses advected from the Adriatic Sea toward the storm centre play an important role in feeding the storm with latent heat. A map showing this behaviour is reported in the discussion on this paper with Reviewer #1.

We added a comment about this point in Section "4 Discussion and conclusions" writing: "The advection of convection from the sea to the land was important in most case studies considered in this paper, and we can conclude that it plays a fundamental role. There are cases, however, when it is less important, as for the severe and localized storm that occurred in NE Italy on 12 September 2012 (Manzato et al., 2014). For this case, the storm developed and evolved over land, and the difference between the precipitation fields of the CNTRL and F3HA6 is confined inland, over NE Italy, and it is larger than 40 mm (see the discussion of this paper for the map of the precipitation difference between CNTL and F3HA6; Federico et al., 2016)."

Line 511: ". . . improvement in some statistical scores, . . .";

-Corrected.

Line 519: please rephrase into ". . . the performance of the precipitation forecast is clearly dependent on the type of event . . .";

-Rephrased.

Figure 3: apparently, the maximum threshold of 90 mm is too small, thus the peak in simulated rainfall cannot be clearly estimated; please, could you add the information about the maximum precipitation simulated by the model at least in the text?

-The Figure 3 is Figure 4 in the revised paper. We added this information in the Figure 4 caption (the maximum value is 320 mm in Southern Italy; over NE Italy the maximum simulated value is 132 mm). Also, we will add the largest value observed in the text, when commenting Figure 4b. We wrote: "The largest precipitation recorded in NE Italy is 141 mm (13.54E, 45.85N), while more than 200 mm are reported in two stations in Southern Italy (15.84E, 40.31N; 207 mm) and (15.98E, 40.16N; 220 mm)."

Reviewer #2

Overall evaluation: The paper presents the evaluation of a lightning data assimilation, implemented in the RAMS model. Overall, the manuscript is well written and the methodology and results are well discussed relative to the available international lit- erature. The subject is of high interest. I suggest acceptance of the manuscript, sub- ject to some minor comments and technical corrections, which are summarized in the following.

Minor comments

1. L64-76: I believe that the three paragraphs could be merged in one, as they all present briefly examples of lightning data assimilation studies.

   - Done.

2. I suggest that Table 1 is removed from the manuscript. Instead of presenting this detailed information on the domain configuration of RAMS, it could be of more usefulness for the reader to add a simple plot showing the domains or stick to the two figures that are already referenced for giving an overview on domains. The respective description of the domain configuration in L120-126 can remain as is.

-Thank you for this comment, we followed your suggestion including a new Figure 1, showing the domains. Table 1 was removed from the paper and the Figure 1 caption includes some details on the domains.

3. L123: It would be also proper to include model top in hPa.

- In the RAMS model, the model top is fixed in z, the height respect to the sea level. Pressure varies on the model top surface, so we gave an estimation of the model top in hPa (40 hPa), from the model output.

4. L151-152: I believe that simply referring to cloud-to- ground (CG) and intra-cloud (IC) lightning is enough, instead of giving the information currently shown in the parentheses.

-Changed according to the comment.

5. L223-227: This particular part of the manuscript presents a result of the study. Hence, it can be removed from Sec. 2, that is devoted to methodology. It can be moved to the Results section, at the appropriate place.

- This part has been moved in the Results section.

6. L259-L263: I think that this paragraph could be removed as it reports information that is most probably well known to the interested readers. I leave it up to the authors to decide whether it should be removed or stay.

-We agree with reviewer that this part of the paper could be removed because it presents basic definition well known to the large part of readers. Nevertheless, in other papers, we found that reviewers asked for this explanation about the scores and, in the doubt, we maintained the paragraph.

7. L290-293: Please rewrite this part of the manuscript in a more clear way.

Thank you for noting this point. We clarified: "During SOP1, several upper level troughs extended from the Northern and Central Europe toward the Mediterranean Basin or entered in the Basin as deep trough. Few of them developed a cut-off low at 500 hPa; the interaction between the upper level troughs and the orography of the Alps generated a low pressure pattern at the surface in Northern Italy, and usually the whole system moved along the Italian peninsula. The 27 October 2012 case study, also referred as IOP16a, belongs to this class of events, but it eventually evolved in a cut-off at 500 hPa on 28-29 October (IOP16c)."

**Technical corrections:**

1. L1: "short-range"

-Done

2. L18: "set up"

-Done

3. L21, L78, L95, L190: "that occurred"

-Done

4. L21: "which were..."

-Done

5. L23, L244, L313 and throughout the entire manuscript: "rain gauges"

-Ok, corrected throughout the paper.

6. L23, L95: "target region" or "target area"

-"Target region".

7. L73: "presented" or "introduced" could be a better choice for this sentence.

-We used "introduced".

8. L75: "performed" instead of "made".

- Done.

9. L131: The correct terminology is "WRF single-moment six-class microphysics scheme"

- Corrected.

10. L174, L322: "setup".

- Done.

11. L174: "trial and error analysis..."

-Corrected

12. L244: The abbreviation QPF (Quantitative Precipitation Forecast) has not been previously defined, should I have not missed it while reading the manuscript.

-Corrected. We apologize for the mistake.

13. L249: "with" instead of "being".

-Done.

14. L267, L484: "competing" instead of "competitor".

- Done.

15. L298-290: Correct the term to "troughs" (it is now written as "through").

- Thank you for helping with this error.

16. L290: "cut-off low".

- Done.

17. L495: "WRF-LTNGDA"

-Ok.

**Relevant changes to the paper**

- A new figure showing the domains of the model (Figure 1).
- A new Appendix (Appendix A) showing the methodology used for the statistical test.
- The text was changed according to the comments of the reviewers (see the marked-up manuscript version of the paper).

[revised manuscript text omitted]

c)

[Figure]

b)

[Figure]

c)                                          d)